# Community Health Worker-Led Cardiovascular Disease Risk Screening and Referral for Care and Further Management in Rural and Urban Communities in Rwanda

**DOI:** 10.3390/ijerph20095641

**Published:** 2023-04-25

**Authors:** Jean Berchmans Niyibizi, Seleman Ntawuyirushintege, Jean Pierre Nganabashaka, Ghislaine Umwali, David Tumusiime, Evariste Ntaganda, Stephen Rulisa, Charlotte Munganyinka Bavuma

**Affiliations:** 1College of Medicine and Health Sciences, University of Rwanda, Kigali 4285, Rwanda; ntawuseleman@nursph.org (S.N.); nganajp@gmail.com (J.P.N.); umwalig@yahoo.fr (G.U.); dktumusiime@gmail.com (D.T.); s.rulisa@gmail.com (S.R.); charlottebavuma5@gmail.com (C.M.B.); 2Global Public Health, Karolinska Institute, 171 77 Stockholm, Sweden; 3Non-Communicable Diseases Division, Rwanda Biomedical Center, Kigali 7162, Rwanda; crisntaganda@gmail.com

**Keywords:** community health workers, cardiovascular diseases risk screening, Rwanda

## Abstract

Cardiovascular disease (CVD) is a global health issue. Low- and middle-income countries (LMICs) are facing early CVD-related morbidity. Early diagnosis and treatment are an effective strategy to tackle CVD. The aim of this study was to assess the ability of community health workers (CHWs) to screen and identify persons with high risks of CVD in the communities, using a body mass index (BMI)-based CVD risk assessment tool, and to refer them to the health facility for care and follow-up. This was an action research study conducted in rural and urban communities, conveniently sampled in Rwanda. Five villages were randomly selected from each community, and one CHW per each selected village was identified and trained to conduct CVD risk screening using a BMI-based CVD risk screening tool. Each CHW was assigned to screen 100 fellow community members (CMs) for CVD risk and to refer those with CVD risk scores ≥10 (either moderate or high CVD risk) to a health facility for care and further management. Descriptive statistics with Pearson’s chi-square test were used to assess any differences between rural and urban study participants vis-à-vis the key studied variables. Spearman’s rank coefficient and Cohen’s Kappa coefficient were mainly used to compare the CVD risk scoring from the CHWs with the CVD risk scoring from the nurses. Community members aged 35 to 74 years were included in the study. The participation rates were 99.6% and 99.4% in rural and urban communities, respectively, with female predominance (57.8% vs. 55.3% for rural and urban, *p*-value: 0.426). Of the participants screened, 7.4% had a high CVD risk (≥20%), with predominance in the rural community compared to the urban community (8.0% vs. 6.8%, *p*-value: 0.111). Furthermore, the prevalence of moderate or high CVD risk (≥10%) was higher in the rural community than in the urban community (26.7% vs. 21.1%, *p*-value: 0.111). There was a strong positive correlation between CHW-based CVD risk scoring and nurse-based CVD risk scoring in both rural and urban communities, 0.6215 (*p*-value < 0.001) vs. 0.7308 (*p*-value = 0.005). In regard to CVD risk characterization, the observed agreement to both the CHW-generated 10-year CVD risk assessment and the nurse-generated 10-year CVD risk assessment was characterized as “fair” in both rural and urban areas at 41.6% with the kappa statistic of 0.3275 (*p*-value < 001) and 43.2% with kappa statistic of 0.3229 (*p*-value =0.057), respectively. In Rwanda, CHWs can screen their fellow CMs for CVD risk and link those with high CVD risk to the healthcare facility for care and follow-up. CHWs could contribute to the prevention of CVDs through early diagnosis and early treatment at the bottom of the health system.

## 1. Introduction

Cardiovascular disease is a global health issue. In 2013, 17 million people died from cardiovascular disease (CVD), the second leading cause of death in the globe, and over 80% of these incidents occurred in low- and middle-income countries (LMICs) [1]. With these statistics, it is obvious that cardiovascular disease significantly increases the burden of noncommunicable diseases (NCDs) in LMICs [2,3]. Not only do CVDs burden LMICs with morbidity and mortality, but they also have an effect on human resources and health services [3]. If no appropriate action is taken, the burden of the CVD pandemic will keep increasing in LMICs and become irrepressible as these countries go through an epidemiological transition and as the proportion of the elderly in populations rises [3]. To reverse the growing CVD burden in LMICs, robust and evidence-based prevention strategies must be developed [4]. Early identification and treatment of those with high CVD risk may lessen the burden brought on by CVD [5].

The proportion of individuals with CVD risk varies from setting to setting. One study revealed that the proportion of people aged 40 to 64 years assessed to be at a higher than 20% risk of CVD varied from less than 1% in Uganda to more than 16% in Egypt when applied to data from 79 countries (mostly LMICs) [6]. A recent secondary analysis of data from the World Health Organization (WHO)-STEPS population-based survey on non-communicable diseases (NCDs) risk factors conducted from November 2012 to 2013 in Rwanda revealed that 4.4% of the Rwandan population have a high 10-year CVD risk (10-years CVD risk score ≥20.0%) [5]. The primary analysis of the same survey had previously revealed that CVD risk factors are prevalent among populations of Rwanda; for instance, 99.1% of the population reported low fruit and vegetable intake; 19.1% of men and 7.1% of women smoked; 30.0% of men and 17.0% of women were heavy drinkers of alcohol at the time of the interview [7]. The prevalence of diabetes in rural and urban areas is 7.5% and 9.7%, respectively [8]. The overall prevalence of hypertension was 15.3% [9]. Furthermore, the recent Rwanda vital statistics report for 2020 data showed that 34.7% of all usable causes of mortality were from NCDs, including CVD [10].

Fortunately, reducing risk factors like tobacco use, poor diets, and excessive alcohol intake, to name a few, can avert many premature deaths from CVD [3]. In addition to other multi-sectoral community-based interventions focusing on CVD risk factors in the general population, the strategy of identifying and managing people at high risk of CVD is essential for CVD prevention and control [11]. Absolute CVD risk evaluation is now encouraged by international guidelines when evaluating and managing CVD risk factors [12]. However, this intervention is challenged by insufficient numbers of health care professionals in LMICs [3]. LMICs need to develop new strategies, such as task shifting or sharing, to deal with the strain placed on their current human resources [3]. Task shifting from licensed doctors and nurses to CHWs can help with the prevention, control, and treatment of chronic NCDs, including CVDs [3]. The CHW-led CVD risk screening is perceived handy in LMICs where there are frequently severe deficits of medical professionals [2]. In low-resource settings, using CHWs for this screening would free up trained health professionals to perform duties that require extensive formal and professional training [2]. The workload of doctors and professional nurses who are presently in charge of the prevention, management, and control of NCDs, including CVDs, is reduced by assigning some roles and responsibilities to CHWs [3]. Moreover, The CHW-led CVD risk screening using mobile technology was proved to be very cost-effective or even cost-saving compared to the usual clinic-based screening [13].

In developed countries, it is common to find CVD risk calculators integrated into software; however, this is rare in low-income countries [12]. Various studies have recommended using mobile health (mHealth) to screen for CVD risk (factors) for identifying and referring individuals with high risk for pharmacological intervention [14]. To date, a number of risk prediction models have been created globally to assess the overall risk of CVDs [15]. Among these tools, some are laboratory-based (algorithm) and costly, whereas others are non-laboratory-based (algorithm) and cost-effective. Different studies from different settings proved that there is concordance between laboratory-based and non-laboratory-based (Body Mass Index (BMI)-based) tools in 10-years CVD risk prediction and characterization [5,15,16,17,18].Thus, authors of these studies, in their conclusions, recommended that non-laboratory-based tools be used in low-resource settings for CVD risk screening. The Framingham non-laboratory-based tool (termed BMI-based tool in this paper) which is used to determine sex-specific CVD risk prediction with the use of inputs of eight CVD risk factors—age, body mass index, systolic blood pressure (SBP), blood pressure (BP) treatment, smoking, and diabetes—was found to be as accurate as a Framingham laboratory-based tool [5]. The two tools use the same inputs except that BMI is replaced with both Total Cholesterol (T-C) and high dose lipoprotein cholesterol (HDL_C) in the Framingham laboratory-based equation [5]. The comparability in CVD risk prediction and characterization between two tools among the Rwandan population was validated using data from WHO STEPs NCDs risk factors survey conducted in Rwanda from November 2012 to March 2013 [5]. Of note, the study conducted among the population of Rwanda proved that BMI-based algorithms detects more people at moderate and high risk than lipid-based algorithms regardless of the sex [5]. The same study recommended a CHW-led CVD risk screening and referral study in Rwandan using the non-laboratory (BMI)-based tool, and this agrees with other authors who recommended the community level CVD risk screening using the non-laboratory tool rather than the laboratory tool in low-resource settings [16,17].

CVD risk screening tools were tested in LMICs, and the results were promising. For instance, one of the previous studies conducted in four countries—Bangladesh, Guatemala, Mexico, and South Africa—proved that trained CHWs successfully used non-laboratory tools to screen for cardiovascular disease risk and produced results that were highly congruent with those produced by trained health professionals (physicians and nurses) [3]. A recent study conducted in one of the low-income countries, Nepal, proved that community health volunteers (CHVs)-led CVD risk screening works and reported a concordance between CVD risk scores generated by CHVs and CVD risk scores generated by doctors [19]. It was also proved in regions of rural Kenya that CHWs can use mHealth tool to screen for CVD risk factors [14].

When determining disease risk factors and the relative efficacy of treatments, including nutrition and lifestyle changes, ethnicity, culture, and context are important considerations [20]. To the best of our knowledge, no study has been conducted yet in Rwanda to test or pilot the CHW-led CVD risk screening using the non-laboratory-based tool as a basis of evidence-based risk in this approach or further research. Importantly, a recent qualitative study reported that rural and urban community members in Rwanda prefer to be screened by CHWs for CVD risk [4]. Thus, the CHW-led CVD risk screening and referral study was warranted in the context of Rwanda.

The aim of this study was dual-fold; it initially aimed at training CHWs to screen and identify high CVD risk individuals in the communities using a CVD risk assessment mobile app and refer them to the health system for care and follow-up thereafter. It also aimed at comparing the accuracy of the CHWs in CVD risk scores against those generated by health care providers at health care facilities using the same tool.

## 2. Materials and Methods

### 2.1. Context of the Study

The implementation of this study adapted and was guided by the protocol titled: “Implementing and Evaluating Community Health Worker-Led Cardiovascular Disease Risk Screening Intervention in Sub-Saharan Africa Communities: A Participatory Implementation Research Protocol”, published elsewhere [21]. Due to differences among different settings (Ethiopia, Malawi, South Africa, and Rwanda) regarding aspects of the populations’ lives, some aspects of the initial protocol were slightly modified to contextualize the implementation of the protocol in Rwanda, and thus there is a need to describe the methods of this study in Rwanda in the present paper.

### 2.2. Study Design

This study was an action research study, whereby CHWs were trained to screen and identify high CVD risk individuals in the communities using a BMI-based CVD risk assessment tool programmed in the KoboCollect App installed in the smart mobile phone, and refer them to a health facility for care and follow-up.

### 2.3. Settings, Community Health Worker Selection, and Participants

This study was conducted in rural and urban communities conveniently sampled from the Burera district, Northern Province, and Gasabo district, City of Kigali. As reported in our previous (qualitative) study that investigated how community members perceive themselves and what communication strategies they prefer in regard to CVD risk among rural and urban residents [4], the term ‘community’ refers, in this study, to the catchment area of a health center (HC) providing NCDs (including CVD) services.

In each community, five villages (i.e., 10 villages for both sites) were randomly selected to be involved in the study. The village is the lowest local government-administrative unit in Rwanda, as illustrated elsewhere [22]. In each selected village, one community health worker (CHW) was identified, in collaboration with the health center staff, to be trained and deployed to his/her village to screen his/her fellow community members (CMs) for CVD risk and refer those with either moderate or high CVD risk (≥10 CVD risk score) to a health facility for further assessment, treatment and management.

The eligibility criteria for local community members to be part of this study included: age bracket of 35–74, having health insurance, understanding and speaking local language (Kinyarwanda), no previous known history of CVD (such as heart attack, myocardial infarction, heart failure, stroke, angina etc.), no skin disease, not pregnant, without disease/condition that could interfere with data collection such as severe mental health conditions (per CHW judgement) and voluntarily consenting to be part of the study.

### 2.4. Developing the CHW Training Manual and Training the CHWs

Prior to the training, on 6–12 March 2022, a training workshop for the pilot study was conducted. During this training, two CHWs from two different villages that were not initially involved in the study were invited for the pilot stage of the study. At that time, CHWs were trained in taking the measurements needed in the Framingham non-laboratory-based CVD risk assessment algorithm for computing individual 10-year CVD risk scores. This algorithm was termed a BMI-based algorithm in our previous work that investigated the comparability of lipid-based (laboratory-based) and BMI-based CVD risk scores [5]. The inputs and details on features of the BMI-based algorithm are published elsewhere [5].

During the training workshop, all CHWs were given theoretical training on how to take measurements needed for determining CVD risk score using a BMI-based algorithm programmed into the KoboCollect App installed in their respective smart mobile phones. Afterwards, only two CHWs from two villages selected for the pilot study per each study site practiced until they knew how to take measurements and use the BMI-based algorithm that was programmed into their smart mobile phones. After two days of training, two CHWs from villages selected for the pilot study were deployed in their respective villages to screen 10 community members for CVD risk within two days. The CMs diagnosed with individual CVD risk score ≥10% (at moderate or high CVD risk) were requested to go to health centers within eight days. After two days of the piloting study, all CHWs and project research teams met at the training workshop site, and two CHWs who did pilot study shared their experience which informed the development of the CHW training manual and the preparation of the training of CHWs from the villages sampled for the study. One of the key lessons learned from the pilot study is that, at the urban site, community members with private health insurance (government and private servants) did not accept being referred to health centers; they preferred to go to private clinics or provincial or national referral hospitals, as they accept their health insurance. At the rural site, community members preferred to go to the health posts located nearby. This resulted in adapting the referral procedures to allow the screened community members diagnosed with either moderate or high CVD risk to be referred to their preferred health facilities (details on the training workshop for the pilot study are available in an unpublished report).

After the pilot study, two consecutive workshops to develop a CHW training manual on CVD risk screening and referral were organized on 9–13 May and 23–28 May 2022, and were attended by the project research team, the health center, and Rwanda Biomedical Center (NCDs division) staff. The training manual consisted of the following topics: (i) introduction to research and research ethics; (ii) general knowledge on CVDs, epidemiology of CVDs at global and national levels, a definition of CVD risk and its estimation; (iii) taking measurements of variables to estimate global CVD risk; (iv) education regarding abnormal measurements and first aid where necessary.

A three-day training, 28–30 September 2022, was organized to train five CHWs per each site (i.e., 10 CHWs for two sites). The training covered five topics of the training manual and questionnaire programmed in the KoboCollect App in the CHWs smart mobile phones. The questionnaire consisted of questions about socio-demographic characteristics as well as self-perception and healthcare-seeking behavior in relation to CVD (risk) and individual 10-year CVD risk scores (Appendix A). The questions to acquire inputs for the BMI-based algorithm included sex, age, height, weight, systolic blood pressure (SBP), taking medication for high blood pressure or not, smoking or not, and having been diagnosed with diabetes or not. If the participant responded that they had not been diagnosed with diabetes, the CHWs asked the participant if he/she had taken something (except water) to eat or drink on the same day and his/her blood glucose was measured with the glucometer. The participant was considered to have diabetes if his/her blood glucose was ≥126 mg/dL (≥7 mmol/dL) or ≥200 mg/dL (≥11.1 mmol/dL) for fasting blood glucose or random blood glucose, respectively. Otherwise, the participant was considered not to have diabetes. The questionnaire was programmed in such a way that the BMI was automatically calculated from the records of weight (kilograms) and height (meters) taken by CHWs and uploaded in the KoboCollect App installed in their smart mobile phones.

CHWs were trained how to use portable balance, a tap meter, a blood pressure monitor, and a glucometer to take measurements of weight, height, SBP, diastolic blood pressure (DBP), and blood glucose, respectively. SBP and DBP were taken three times and it was the average between the second and the third readings of SBP used in the BMI-based algorithm that was used to determine the individual CVD risk score. In this study, tape meter was used to measure waist circumference and hip circumference that were recorded for each study participant.

### 2.5. Sample Size and Sampling Study Participants

This study was designed in such a way that each CHW should screen 100 persons residing in the village he/she serves; this meant that the sample size was 1000 (500 per study site). After training CHWs, a two-day training workshop was organized to repeat the administration of the questionnaire, taking of measurements, and sampling of the study participants. In this training workshop, a CHW and executive secretary of the cell— from which a village was sampled for the study—used the village registry to identify households that had at least one person eligible for this study, referring to the eligibility criteria provided earlier. During this exercise, we noticed that, except for one village from the rural site, all villages had less than 100 households that were eligible for the study, but there were some households that had more than one person eligible for the study. During this training workshop, one CHW at the urban site reported that her village would not have the sample size required per village; thus, this CHW was allowed to recruit study participants from a neighboring village.

### 2.6. Data Collection

Data for this study were collected in three phases by three different categories of data collectors. The first phase of data collection was conducted by CHWs at the village level and it targeted community members fulfilling the eligibility criteria mentioned under the sub-section of settings, community health worker selection, and participants. During this phase, CHWs used a questionnaire (Appendix A) programmed in the KoboCollect App that was installed in their smart mobile phones. Data collection for the CHW-led CVD risk screening and referral for care started on 12 October 2022 and ended on 20 November 2022, even though it had been planned to end on 12 November 2022. The CHW-led CVD risk screening and referral for care was extended to 20 November 2022 (one week more), because there were some CHWs who had not yet reached their target (100 community members per village) on 12 October 2022. After ensuring that CVD risk screening was completed at the community level, the three weeks from 21 November to 3 December 2022 were considered to be the period for health facilities to receive community members diagnosed with moderate or high CVD risk. It is important to know that community members with low CVD risk (CVD risk score < 10) but with abnormal measurements for blood glucose and blood pressure were referred to health facilities. However, community members with low CVD risk (CVD risk score < 10) but with abnormal measurement for blood glucose and blood pressure were not considered in the analysis of this study.

When we contacted health providers (NCD nurses and laboratory technician) at health centers on 4 December 2022 and asked them if community members referred by CHWs in the context of the study were still coming, they confirmed that they were still coming. Thus, one week was added to wait for additional community members referred by CHWs to come to the health facility. Therefore, the period of receiving community members referred by CHWs to the health care facility was extended to 11 December 2022 when the health center staff reported they were no longer receiving the community members referred for this study.

The second phase of data collection was conducted by health center staff and targeted community members screened and referred by CHWs to health facilities because they had been diagnosed with either moderate or high CVD risk (CVD risk score ≥ 10). Health center staff used the same questionnaire (Appendix A) as the one used by CHWs; this questionnaire was also programmed in the KoboCollect App that was installed in their (nurses’) smart phones. However, nurses collected only data that were inputs of BMI-based CVD risk scores. This data collection was conducted during the period of 12 October to 11 December 2022.

The third phase of data collection was conducted by hired data collectors and targeted all community members screened for CVD risk, diagnosed with either moderate or high CVD risk, and who had accepted to be referred to the health facility for care and further follow-up. The study participants from this phase were obtained from the referral log sheets of the CHWs. This phase mainly aimed to know how the referred community members who had attended the health facilities appreciated the services received at the health facilities. It also aimed to learn why community members diagnosed with either moderate or high CVD risk and accepted to be referred to health facility did not go to the health facility. The details on the questionnaire used during this phase is available in Appendix B. The data were collected through phone interviews and recorded on papers. After paper-based data collection, the questionnaire was programmed into KoboCollect App and deployed via a web-based link used by data collectors to enter data that were initially recorded on papers. The data collection for phase three was conducted during the period of 6–27 February 2023.

### 2.7. Data Analysis

Prior analysis data were cleaned by removing duplicates in each dataset (dataset for all community members screened by CHWs, dataset for community members screened by health center staff, and dataset of community members screened by CHWs and referred to health care facilities because of their moderate or high CVD risk). Normality of continuous variables was assessed using both box plot and histogram. Descriptive statistics were presented using the median (25th–75th percentiles) for continuous variables, as the latter were asymmetric, and absolute frequencies and percentages for the categorical variables. Pearson’s Chi-squared (χ^2^) and Wilcoxon rank-sum test were used for categorical variables and continuous variables, respectively, to compare rural and urban areas in terms of different variables investigated in this study.

Spearman’s rank correlation coefficient was used to assess the correlation between the individual CVD risk scores generated by CHWs and those generated by nurses, and scatter plots were used to present the correlation between them. CVD risk was categorized as <10% low risk, 10–20% moderate risk, and ≥20% high risk, as it was in previous studies [5,23]. Statistical testing procedures for Cohen’s Kappa were conducted to test the level of agreement between CHW-based characterization and self-characterization of study participants into three categories of a 10-year CVD risk (low 10-year CVD risk, moderate 10-year CVD risk, and high 10-year CVD risk). The kappa statistic was also used to assess the agreement between CHWs and nurses in characterizing study participants into the abovementioned three CVD risk categories. The latter analysis only concerned the study participants diagnosed with either moderate or high CVD risk (individual CVD risk score ≥10). Stata 13 (Stata Corporation, College Station, TX, USA) was used for all statistical tests, with a significance level of 5%.

### 2.8. Ethical Considerations

The protocol for this research was reviewed and approved by the Rwanda Ministry of Health ethics review committee known as the Rwanda National Ethics Committee (RNEC) (No. 807/RNEC/2021). All study participants provided a written informed consent before getting involved in the study.

## 3. Results

### 3.1. Characteristics of CHWs Volunteered in the Study

In this study, 60.0% of CHWs from both rural and urban sites who volunteered were females; 100% were female in urban areas, whereas in rural areas the participants were predominantly males, 80.0%. The mean age of CHWs who volunteered in this study were 39.6 years. The youngest CHWs was aged 32 years in both rural and urban areas, while the oldest in rural and urban areas were aged 45 years and 49 years, respectively. The proportion of CHWs for whom the highest level of education was three years of secondary education, and the proportion of those for whom the highest level of education was the secondary education was the same, 40.0%, meaning that 80.0% of CHWs either did three years of secondary education or completed secondary education. This distribution was the same in rural and urban communities. The CHW with the lowest level of education (Five years of primary education, P5) and the CHW with the highest level of education (Bachelor, A0) were recorded in rural and urban areas, respectively.

### 3.2. Characteristics of Study Participants

The rate of study participation is almost 100% and the same in both rural community and urban community, 99.6% (498/500) vs. 99.4% (497/500), and the overall majority of the study participants were women, 56.6%. This dominance of women in study participation was observed and is 57.8% vs. 55.3% for the rural area and urban area, respectively. The median age (25th–75th percentiles) of study participants was significantly higher in the rural area than in the urban area, 48.5 (40–59) vs. 44 (39–53), with *p*-value < 0.001. Overall, majority of the study participants were married or cohabitants, 81.41%, and the same observation was made in the rural and urban areas, where the biggest proportion of study participants were married/cohabiting, 92.6% vs. 70.2%. As indicated in Table 1, the difference in the proportion of study participants between rural areas and urban areas vis-à-vis marital status is significant (*p*-value < 0.001). For the educational level, the proportion of study participants with lower levels of education (no formal education or incomplete primary education) was higher in rural areas than in urban areas, whereas this pattern reverses for the study participants with at least the completion of primary education. This difference in level of education between rural and urban areas is statistically significant, *p*-value < 0.001 (Table 1). Most study participants were of the Ubudehe (socio-economic status in Rwanda) category 3 for the overall study participants, 47.9%, in the rural community, 44.4%, and in the urban community, 51.5%. The difference in study participants between the rural and urban communities in terms of wealth status was statistically significant, *p*-value <0.001 (Table 1).

As indicated in Table 1, the majority of study participants used community-based health insurance schemes, 92.4%, and this was the case for the study participants in the rural community (96.6%) and in the urban community (88.1%); the difference between rural and urban areas in terms the type of health insurances used by community members is significant, *p*-value < 0.001 (Table 1). Table 1 indicates that the largest proportion of study participants were of normal BMI, 57.1%. Surprisingly, many study participants in rural area were of normal BMI (71.6%), whereas the majority of study participants, half of them (50.1%), in the urban area were either obese or overweight, and the difference between rural and urban areas is statistically significant (*p* < 0.001) (Table 1). As indicated in Table 1, the proportion of study participants with high CVD risk (≥20%) is 7.4% for overall study participants and is slightly higher in the rural area than in the urban area, 8.0% vs. 6.8%, respectively (Table 1). The point prevalence of study participants diagnosed with either moderate or high CVD risk (≥10%) was 23.9%, 26.7%, and 21.1% for the overall study participants, rural study participants, and urban study participants, respectively (Table 1).

This study revealed that community members with high 10-year CVD risk underestimate their risk; as indicated in Table 2, 32.5% and 35.3% of study participants diagnosed with high CVD risk in rural area and urban area, respectively, think that they are not at risk of heart attack or any heart disease at all (Table 2). It is important to note that, as indicated in Table 2, the proportion of study participants with high 10-year CVD risk that were very willing to attend health facilities for medical check-ups to rule out any heart disease was higher in the urban than the rural areas, 44.1% vs. 5.0%. Another indicator of underestimation by community members of their CVD risk was that more than a quarter, 35.0%, and more than a half, 55.9%, of study participants diagnosed with high CVD risk in rural and urban areas, respectively, perceived themselves to be at low CVD risk (Table 2).

Further investigation of the agreement between the CHW-based characterization and self-characterization of study participants into three categories of a 10-year CVD risk (low 10-year CVD risk, moderate 10-year CVD risk, and high 10-year CVD risk) indicated that the observed agreement was slight and equaled 48.4% with a Kappa statistic of 0.02, not statistically significant (*p*-value = 0.239) for overall study participants. This agreement was also slight in the rural area, 48.4%, with a Kappa statistic of 0.04, which was not statistically significant (*p*-value = 0.128). In the urban area, it was found that the agreement between the CHW-based characterization and self-characterization for study participants into three categories of a 10-year CVD risk (low 10-year CVD risk, moderate 10-year CVD risk, and high 10-year CVD risk) was worse and equaled 48.5% with negative value of Kappa statistic (−0.009) with *p*-value = 0.617.

### 3.3. Agreement between CVD Risk Scores Determined by CHWs and CVD Risk Scores Determined by Nurses

In this study, less than half, 40.5% (69/171), of study participants screened by CHWs, diagnosed with either moderate or high CVD risk (CVD risk score ≥10%) and given a referral note attended a health facility. Almost half, 49.5% (46/93), and less than a quarter, 16.7% (13/78), of study participants diagnosed with either moderate or high CVD risk (CVD risk score ≥10%) who were given referral slips attended the health facility in rural area and urban area, respectively. This study also found that there is a strong positive correlation between CVD risk scores measured by CHWs and CVD risk scores measured by nurses in general study participants, rural study participants, and urban study participants (see Figure 1), as evidenced by their respective Spearman rank correlation coefficients: 0.6504 (*p*-value < 0.001), 0.6215 (*p*-value < 0.001) and 0.7308 (*p*-value = 0.005).

The observed agreement between the CHW-generated 10-year CVD risk scores and the nurse-generated 10-year CVD risk scores was 42.6% with a kappa statistic of 0.3183 (*p*-value <0.001), 41.6% with kappa statistic of 0.3275 (*p*-value < 001) and 43.2% with kappa statistic of 0.3229 (*p*-value = 0.057) for overall study participants, rural study participants, and urban study participants; respectively.

### 3.4. Outcomes and Experiences on Referral

During the phase of collecting data on referral feedback, data were collected through phone interviews, the response rate was almost 100% as 170/171 (99.4%) of study participants diagnosed with either moderate or high CVD risk and referred to a health facility by CHWs were interviewed upon their acceptance. As indicated in Table 3, 68.8% of overall study participants referred to a health facility because of either moderate or high CVD risk attended the health facility. Compliance with referrals given by CHWs was significantly higher in rural areas than in urban areas, 76.6% vs. 59.7% (*p*-value = 0.020). The majority of community members referred to health facilities, 95.7%, attended health centers, and there was no difference between rural and urban areas (Table 3). Overall, 75.2% of study participants who attended health facilities accepted having received health care at a health facility, and more than a quarter of study participants who attended health facilities in both rural (71.8%) and urban areas (80.4%) accepted having received health care (Table 3).

Over half, 54.7%, of overall study participants who were referred to health facilities and who complied with the referral provided by CHWs, reported to have been followed up by the health facility; furthermore, it was reported that over a half of study participants who were referred to health facilities and who complied with the provided referral in both rural and urban areas, 52.1% and 58.7%, respectively, confirmed that they were followed up with by the health facility (Table 3). As indicated in the Table 3, 30.8% of study participants who were referred to health facilities and who complied with the referral, were diagnosed with either hypertension, diabetes, or both and were put on medication for the first time due to those diseases. Specifically, 22.5% and 43.5% of study participants referred to health facilities and who complied with the referrals in rural and urban areas, respectively, were diagnosed with the aforementioned diseases and were put on medication for the first time (Table 3). In this study, it was found that 90.5% of study participants who were referred to health facilities and who complied with the referral of CHWs were at least satisfied with the services they received from the health facility. Based on the same research question, it was also highlighted that 91.4% and 89.1% of study participants referred to health facilities and who complied with the referrals of CHW in rural and urban areas, respectively, were at least satisfied with the services they had received from health facility (Table 3).

In the present study, as illustrated in Figure 2, it was found that financial constraints, such as lack of user fees, and unavailability to go to health facility were the two main reasons for not having complied with the referral note provided by the CHW. The first reason for not complying with the referral provided by CHWs, in urban areas, was the lack of the time to go to health facility, whereas the reason was lack of money for study participants in rural areas. Surprisingly, 4.6%, and 1.9% of rural study participants and urban study participants, respectively, who were referred to health facility by CHWs but who did not comply with the referral reported lack of trust in CHWs as the reason of not having complied with the referral provided by the CHWs (Figure 2).

## 4. Discussion

In this study, we assessed the potential for CHWs to use mobile-technology with non-laboratory CVD risk screening tool to identify moderate and high CVD risk individuals in the communities and refer them to the health system for care and follow-up thereafter. This study has shown that CHWs can screen their fellow community members for CVD risk using non-laboratory (BMI)-based tool programmed in KoboCollect App installed in the smart mobile phone. CHW-generated CVD risk scores are in correlation with nurse-generated CVD risk scores, meaning that the ability of CHWs to conduct CVD risk screening in their local communities is unquestionable. This study also indicated that compliance with referrals provided by CHWs is promising; however, compliance is very low in the urban community. This study helps to diagnose local community members with diseases that are NCDs/CVDs risk factors and put them on medication for the first time.

This study provided proof that CHWs can screen their fellow community members for (high) CVD risk, and community members accepted being screened by CHWs for (high) CVD risk in Rwanda. This is not surprising, since in one qualitative study recently conducted in Rwanda, community members expressed a need to be screened by CHWs for CVD risk [4]. A strong positive correlation between CVD risk scores measured by CHWs and CVD risk scores measured by nurses is proof that CHWs have the potential to conduct CVD risk screening using mobile-technology. The potential of community health workers to screen community members using mHealth tool was recently assessed and proved in Kenya [14]. In the present study it was found that the observed agreement between the CHW-generated 10-year CVD risk characterization and the nurse-generated 10-year CVD risk characterization for either overall study participants, rural study participants, or urban study participants is fair, as the computed kappa statistics are in range of 0.21–0.40 [19], and statistical significance was observed only for overall study participants and rural study participants. These results are far lower than those found in Nepal where the agreement between CHV-generated 10-year CVD risk characterization and doctor-generated 10-year CVD risk characterization for either overall study participants, rural study participants, or urban study is substantial as the computed kappa statistics were in range of 0.61–0.80 [19], and were all statistically significant.

This study shows that the point prevalence of high CVD risk (≥20%) is 7.4% among overall study participants, which is 1.7 times higher than that found among Rwandan populations in 2013, 4.4% [5]. This increment can be attributed to either one of the two reasons. The first reason may be that, in addition to their self-reported diabetes status, the current study confirmed the study participants’ diabetes status after measuring blood glucose, in case a study participant self-reported to not have diabetes. The second reason may be the change in the Rwandan population in their CVD risk factor profiles. The point prevalence of high CVD risk in Rwanda is higher than that found in Kenya, 0.8% [14]. However, this difference is inconclusive since CVD risk screening tools used in these two settings are different. In Rwanda, this study indicated that the point prevalence of high CVD risk is slightly higher among rural study participants than among urban study participants, 8.0% vs. 6.8%. The cumulative point prevalence of either moderate or high CVD risk (≥10%) is 23.9% among the overall study participants, and it is higher among rural study participants than among urban study participants, 26.7% vs. 21.1%. This is not in agreement with another study investigated on CVD risk and events in 17 low-, middle-, and high-income countries where it was found that urban areas have higher CVD risk scores compared to the rural areas in LMICs [24]; this can be interpreted as meaning that the proportion of community members with high CVD risk is higher in urban than in rural areas.

Even though most of the community members accepted being screened by CHWs for CVD risk, compliance with the referral notes provided by CHWs to go to health facilities for community members diagnosed with either moderate or high CVD risk is low in general and lower among urban community members than among rural community members. Low attendance among community members referred to health facilities because they were at risk for developing a CVD or risk factor of CVD was reported in previous studies in other settings [25,26].

Furthermore, it was found that, in Rwanda, low attendance at health facilities is mainly due to lack of money or user fees and lack of time. Lack of trust in CHWs was also found as one of the reasons for not complying with referrals provided by CHWs. Financial constraint and lack of trust in CHWs were also reported among the reasons for non-attendance at health facilities in South Africa [26]. In the present study, conducted among rural and urban communities in Rwanda, it was found that study participants underestimated their CVD risks. The underestimation of CVD risk among community members was reported in previous studies conducted in other settings [12,27,28]. The misestimation of CVD risk among the community members was recorded in South Africa [25].The CVD risk underestimation recorded in the present study can be linked to poor understanding of the concept of CVD risk, which was reported among rural and urban community members in a previous study conducted in Rwanda [4]. Moreover, this poor understanding of CVD risk can be one of the contributors to low attendance at health facilities, as reported in a study conducted in South Africa [26].The low attendance at heath facilities observed in this study can also be explained by the low proportion of study participants who were very welling to attend a health facility for a medical check-up to rule out any heart disease. The lack of willingness to attend a health facility among community members for a cardiovascular health check was reported by Caroline et al.(2015) in their study investigating patients’ willingness to attend the cardiovascular health checks in primary care [29]. Even though many community members diagnosed with either moderate or high CVD risk and referred to health facilities did not comply with referrals provided by CHWs, it was found that almost all study participants who attended the health facility, 90.5% were satisfied with the services they received from the health facility.

This study helped to identify community members who unknowingly lived with either hypertension or diabetes who were put on medication for the first time. Our study has reported the low number of community members diagnosed with either hypertension or diabetes, because analysis of our study did not consider community members who were referred by CHWs to health facilities because they were diagnosed with either high blood pressure or blood glucose but with low CVD risk (<10 CVD risk score). This is an indicator that Rwandans are unknowingly living with the above-mentioned deadly non-communicable diseases that are silent. The contribution of CHW-led CVD risk screening in diagnosing new hypertension cases was also reported in a previous study conducted in South Africa [26].

This study has four limitations. The first limitation is that two study areas (rural and urban communities) were conveniently selected, and thus the results from this study cannot be generalized to the Rwandan population. However, the findings from this study constitute the baseline of future studies on the CHW-led CVD risk screening and referral in Rwanda and other settings similar to the Rwanda context. The second limitation is that it did not strictly select CHWs as planned in the initial protocol of this study; it was initially planned that 10 CHWs would be trained per each study site and five CHWs would be retained to volunteer in CHW-led CVD risk screening and a referral study by considering the best first five CHWs in post-test training, with more than a 75% score. Due to time and financial constraints, the initial proposal of selecting CHWs was not executed as planned; only five CHWs were trained per each study site, and those who did not perform well were helped until they knew well how to conduct CVD risk screening using mobile technology. Thus, the interpretation of the results regarding the comparison between CHW-generated CVD risk scores and nurse-generated CVD risk scores should be done with caution. The third limitation is that CVD risk scores were calculated using data-for some inputs of the Framingham non-laboratory CVD risk algorithm that were determined from Framingham population, which is different from Rwandan population in various aspects; this implies that the results for CVD risk scores are possibly biased. Thus, we recommend future studies that refer to the WHO HEARTS technical package for CVD management [30]; the latter has the data relevant to the population of similar risk and characteristics of Rwandan population. As recommended in our previous study [5], a 10-year or 5-year cohort study is recommended among the population to determine an accurate CVD risk score. The last but not least limitation is that, due to time and financial constraints, our study did not consider a qualitative phase to explore experience or challenges for the CHW-led CVD risk screening and referral in rural and urban communities of Rwanda. However, during the follow-up data collection on CHW-led CVD risk screening and referral, we used a questionnaire that included questions which could provide superficial information that could help in the design of further studies and scale-up. Thus, we strongly recommend a qualitative study aiming to provide proof of concept on the implementation (research) for the CHW-led CVD risk screening and referral for care and management.

Despite the above-mentioned limitations, the present CVD risk prediction and characterization is more accurate than our previous study [5], because in the present study blood glucose was measured during the time of data collection for study participants who self-reported not living with diabetes. This helps to ensure the diabetes status of the study participants; thus, CVD risk scores calculated in this study are more accurate than our previous study that relied only on community members self-report to know their diabetes status. This practice is in line with the recommendation from the CHW-led CVD risk screening study conducted in Kenya [14], and our previous study that was conducted to assess the comparability between lipid-based and BMI-based CVD risk screening tool [5]. The consideration of rural and urban areas in the design of this study is another strength of this study, as many results from this study are different from rural to urban study sites; this is very important to informing the design of strategies or interventions to prevent CVD (risk) in the context.

## 5. Conclusions

In Rwanda, CHWs can screen their fellow community members for CVD risk (and CVD risk factors such hypertension and diabetes) and link those with high CVD risk or patients to the healthcare system for follow up and proper management. This intervention can help to identify community members unknowingly living with chronic diseases such as hypertension, diabetes etc. and manage them, thereby reducing and preventing CVD risk and event, respectively, among the affected people. To determine whether CHW-led CVD risk screening and referral for care and management is scalable, a further 5-year or 10-year cohort study to determine inputs of CVD risk screening algorithm that are specific to the Rwandan population, implementation science and study on acceptability and feasibility among beneficiaries, CHWs, healthcare workers, and policy makers are needed. Finance and time constraints highlighted to be the issues are to be considered in designing further studies.

## Figures and Tables

**Figure 1 ijerph-20-05641-f001:**
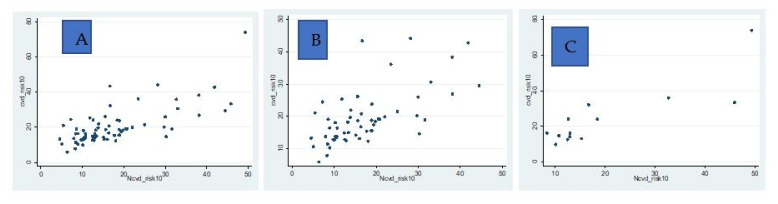
Scatter plots comparing CHW-led CVD risk scoring and nurse-led CVD risk scoring. (**A**): Scatter plot for overall study participants; (**B**): Scatter plot for rural study participants; (**C**): Scatter plot for urban study participants, **cvd_risk10:** CVD risk scores by CHWs and **Ncvd_risk10:** CVD risk score by nurses.

**Figure 2 ijerph-20-05641-f002:**
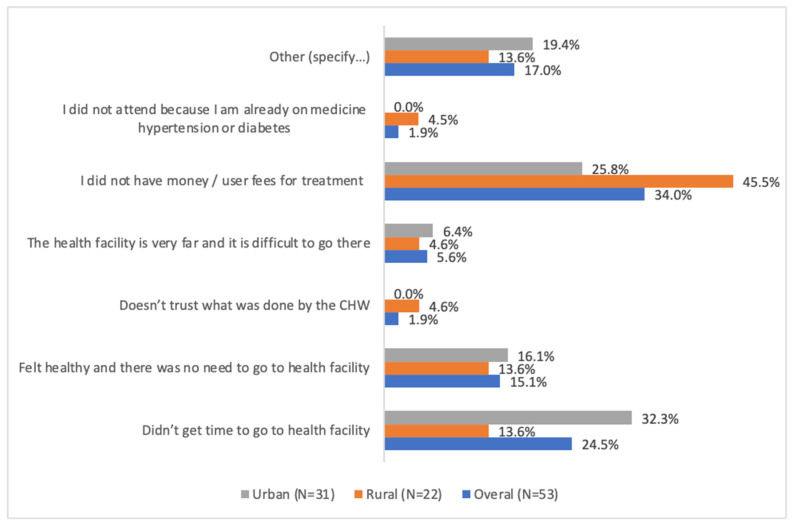
Reasons of non-compliance with the referral provided by CHWs.

**Table 1 ijerph-20-05641-t001:** Socio-demographic, lifestyle and health characteristics of community members screened for CVD risk in rural and urban communities.

Variable		Overall	Rural		Urban		*p*-Value
	*n*	*n*(%)	*n_r_*	*n_r_*(%)	*n_u_*	*n_u_*(%)	
Location	995		498		497		
**Gender**	995		498		497		**0.426**
1. Men		432 (43.4)		210 (42.2)		222(44.6)	
2. Female		563(56.6)		288(57.8)		275(55.3)	
**Age, median (P25–P75)**	995	45 (39–56)	498	48.5(40–59)	497	44(39–53)	<0.001
**Marital status, *n* (%)**	**995**		**498**		**497**		**<0.001**
1. Single		64 (6.4)		6 (1.2)		58 (11.7)	
2. Married/cohabitating		810 (81.4)		461(92.6)		349(70.2)	
3. Divorced/separated		47 (4.7)		4 (0.8)		43(8.7)	
4. Widow/widower		74(7.4)		27(5.4)		47 (9.5)	
**Level of education**	**995**		**498**		**497**		**<0.001**
1. No formal education		248 (24.9)		196(39.4)		52 (10.5)	
2. Primary education not completed		260 (26.1)		174(34.9)		86 (17.3)	
3. Primary education completed		254(25.5)		89 (17.9)		165(33.2)	
4. Secondary education not completed		107 (10.8)		24 (4.8)		83(16.7)	
5. Secondary education completed		97 (9.8)		11 (2.2)		86(17.3)	
6. Post-secondary and above		29 (2.9)		4 (0.8)		25(5.0)	
**Ubudehe category**	**995**		**498**		**497**		**<0.001**
1. Category 1		143(14.4)		103(20.7)		40 (8.1)	
2. Category 2		369 (37.1)		174(34.9)		195(39.2)	
3. Category 3		477(47.9)		221(44.4)		256(51.5)	
4. Category 4		6(0.6)		0 (0.0)		6 (1.2)	
**Health insurance**	**995**		**498**		**497**		**<0.001**
1. Community-based health insurance (*Mutuelle de santé*)		919 (92.4)		481(96.6)		438 (88.1)	
2. RSSB and other private insurance		76 (7.6)		17 (3.4)		59 (11.9)	
**BMI category**	**995**		**498**		**497**		**<0.001**
1. Lean		91(9.1)		57 (11.5)		34 (6.8)	
2. Normal		568 (57.1)		357(71.7)		211(42.5)	
3. Obese and overweight		336(33.8)		84 (16.9)		252(50.7)	
**A 10-year CVD risk score**	995	5.162 (2.792–9.647)	498	5.613 (2.965–10.713)	497	4.799 (2.619–8.636)	0.015
**Categories of 10-year CVD risk**	**995**		**498**		**497**		**0.111**
Low (<10%)		757 (76.08)		365(73.3)		392(78.9)	
Moderate (≥10% and <20%)		164 (16.48)		93 (18.7)		71(14.3)	
High (≥20%)		74 (7.44)		40 (8.0)		34 (6.8)	

**Table 2 ijerph-20-05641-t002:** CVD risk self-perception and healthcare seeking behavior vis-à-vis CVD risk categories.

	Predicted 10-Year CVD Risk					
	Rural			Urban		
Variable	Low (<10%), *n* = 365	Moderate (≥10% and <20%), *n* = 93	High (≥20%), *n* = 40	Low (<10%), *n* = 392	Moderate (≥10% and <20%), *n* = 71	High (≥20%), *n* = 34
**Self-perceived to be at CVD risk**						
1. Not probable	138(37.8)	22 (23.7)	13 (32.5)	108 (27.6)	19 (26.8)	12 (35.3)
2. Somewhat improbable	110(30.1)	41 (44.1)	14 (35.0)	51(13.0)	5 (7.0)	2 (5.9)
3. Neutral	10 (2.7)	2 (2.2)	1 (2.5)	5 (1.3)	0 (0.0)	0 (0.0)
4. Somewhat probable	75 (20.5)	17 (18.3)	9 (22.5)	154(39.3)	29 (40.9)	13 (38.2)
5. Very probable	32(8.8)	11 (11.8)	3(7.5)	74 (18.9)	18 (25.4)	7(20.6)
**Willingness to attend health facility for medical check up**						
1. Not Willing	36 (9.9)	7 (7.5)	5 (12.5)	21 (5.4)	1 (1.4)	2 (5.9)
2. Not Sure yet	99 (27.1)	27 (29.0)	8 (20.0	20 (5.1)	2 (2.8)	0(0.0)
3. Somewhat Willing	38(10.4)	15 (16.1)	8 (20.0)	23 (5.9)	4 (5.6)	1 (2.9)
4. Willing	184(50.4)	38 (40.9)	17 (42.5)	213 (54.3)	39 (54.9)	16 (47.1)
5. Very willing	8 (2.2)	6 (6.5)	2 (5.0)	115 (29.3)	25 (35.2)	15 (44.1)
**Healthcare seeking in last 6 months**						
1.Yes	101(27.7)	27 (29.0)	11 (27.5)	141(36.0)	33 (46.5)	21 (61.8)
2. No	264(72.3)	66 (71.0)	29 (72.5)	251(64.0)	38 (53.5)	13 (38.2)
**CVD risk self-characterization**						
1. Low	192(52.6)	52 (55.9)	14 (35.0)	210 (53.6)	42 (59.2)	19 (55.9)
2. Moderate	156(42.7)	40 (43.0)	17 (42.5)	152 (38.8)	27 (38.0)	11 (32.3)
3. High	17(4.7)	1 (1.1)	9 (22.5)	30 (7.7)	2 (2.8)	4 (11.8)

**Table 3 ijerph-20-05641-t003:** Referral feedback from CHWs to Health Facilities.

Variable	*n*	Overall	Rural		Urban		*p*-Value
		*n* (%)	*n_r_* (%)	*n_r_* (%)	*n_u_* (%)	*n_u_* (%)	
**Complied with referral made by CHW**	**170**		**93**		**77**		**0.020**
1.Yes		117 (68.8)		71 (76.3)		46 (59.7)	
2.No		53 (31.2)		22 (23.7)		31 (46.3)	
**Type of health facility attended n (%)**	**117**		**71**		**46**		**0.402**
1.Health center		112 (95.7)		68 (95.8)		44 (95.6)	
2. Health post		1 (0.9)		0 (0.0)		1 (2.2)	
3. Public hospital		2 (1.7)		1 (1.4)		1 (2.2)	
4. Private clinic		2 (1.7)		2(2.8)		0 (0.0)	
**Accepting of received treatment**	**117**		**71**		**46**		**0.292**
1. Yes		88 (75.2)		51 (71.8)		37 (80.4)	
2. No		29 (24.8)		20 (28.2)		9(19.6)	
**Being followed up by health facility**	**117**		**71**		**46**		
1. Yes		64 (54.7)		37(52.1)		27 (58.7)	
2. No		53 (45.3)		34 (47.9)		19 (41.3)	
**Reason for being initiated on medicine for the first time at health facility**	**117**		**71**		**46**		**0.015**
1. Hypertension		27 (23.1)		15 21.1)		12 (26.1)	
2. Diabetes		5 (4.3)		0 (0.0)		5 (10.9)	
3. Both hypertension and diabetes		4 (3.4)		1 (1.4)		3 (6.5)	
4. Not initiated on the medicine		80 (68.4)		54(76.1)		26 (56.5)	
5. Other (specify…)		1 (0.9)		1 (1.4)		0 (0.0)	
**Level of satisfaction on quality of care received at health facility**	**117**		**71**		**46**		**0.151**
1. Very dissatisfied		3 (2.6)		1(1.4)		2 (4.4)	
2. Dissatisfied		5 (4.3)		4(5.6)		1 (2.2)	
3. Neither satisfied nor dissatisfied		3 (2.6)		1(1.4)		2 (4.4)	
4. Satisfied		62 (53.0)		43(60.6)		19 (41.3)	
5. Very satisfied		44 (37.6)		22(31.0)		22 (47.8)	

## Data Availability

To be compliant with the health leading institutions’ research and data sharing policy in place, the raw data analysed during this study are available from the corresponding author on request.

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
