# Peer review of "Community Health Worker-Led Cardiovascular Disease Risk Screening and Referral for Care and Further Management in Rural and Urban Communities in Rwanda"

_ijerph, 2023, doi:10.3390/ijerph20095641_

Round 1

Reviewer 1 Report

The present study is an interesting and well-designed protocol to promote community health literacy. However, some recommendations would be provided as below:

1.    Abstract section would be condensed and the statistic method could be deleted (Line 24-27) in this section.

2.    Line 99: Please revised “non-laboratory -based (BMI-based) tools” into “non-laboratory -based (Body Mass Index (BMI)-based) tools”.

3.    Line 182: Please revise BMI (Body Mass Index)-based algorithm into BMI-based algorithm.

4.    In Table 1, please add “Gender” at the first variable; revise the second N, Nr and Nu into N (%), Nr (%) and Nu (%) and, place the first P-value (0.426) at its exact area.

5.    Line 313: Revised as “ …CVD risk scores generated by CHWs and those generated by nurses…”

6.    Line 315: Revised as “   , 10% - 20% moderate risk …”

7.    Line 345-6: Please correct the data as below: …majority of the overall study participants are women 56.6 %....

8.    Line 378-9: Please correct the data as below: …in the Table 2, 32.5 % and 35.3% of study participants diagnosed with high CVD risk in rural and urban, respectively, …

9.    Line 382-3: Please correct the data as below: …very willing to attend health facility for medical check-up to rule out any heart disease is higher in urban than in rural, 44.1% vs 5.0%...

10. Line 388-398: Please recheck all the calculations of the agreement in urban, rural and overall study subjects. For example, the crude agreement between the CHW-based characterization and self-characterization should be 48.5% ((210+27+4)/497) for the urban population and 48.4% ((192+40+9)/498) for the rural population.

Author Response

Please find attached the word document providing the feedback to the reviewer 1 comments.

Reviewer 2 Report

1. Selection of an important and relevant topic in CVD control is appreciated.

2. Overall objectives have to be made more clear. a) comparison of lab and non lab based CVD risk scores? it appears like that, but is it comparison of scores with and without blood sugar/diabetes. b) A schematic diagramme can be done to show the different levels 

3. Referral compliance is confusing, in one place it says 50%, other places it says 70% and above.

4.  Type of treatment provided is important as high CVD risk needs an integrated management.

5. WHO has updated CVD risk - using cohorts from LMICS and they may be considered. CVD risk scores can be more relevant when data from populations of similar risk and characteristics and included.  WHO HEARTS technical package

Author Response

Please find attached the word document providing the feedback to the comments from the reviewer 2
